# Characterization of 5356 Aluminum Walls Produced by Wire Arc Additive Manufacturing (WAAM)

**DOI:** 10.3390/ma16072570

**Published:** 2023-03-23

**Authors:** Michal Wieczorowski, Alejandro Pereira, Diego Carou, Bartosz Gapinski, Ignacio Ramírez

**Affiliations:** 1Faculty of Mechanical Engineering, Poznan University of Technology, Piotrowo Street 3, 60-965 Poznan, Poland; 2Escola de Enxeñaría Industrial, Campus Lagoas Marcosende, Universidade de Vigo, 36310 Vigo, Spain; 3Escola de Enxeñaría Aeronáutica e do Espazo, Universidade de Vigo, 32004 Ourense, Spain

**Keywords:** additive manufacturing, WAAM, multi-layered walls, computed tomography, AA5356, tensile testing

## Abstract

Wire arc additive manufacturing (WAAM) is renowned for its high deposition rate, enabling the production of large parts. However, the process has challenges such as porosity formation, residual stresses, and cracking when manufacturing aluminum parts. This study focuses on ana-lyzing the porosity of AA5356 walls manufactured using the WAAM process with the Fronius cold metal transfer system (Wels, Austria). The walls were machined to obtain specimens for tensile testing. The study used computed tomography and the tensile test to analyze the specimens’ porosity and its potential relation to tensile strength. The process parameters analyzed were travel speed, cooling time, and path strategy. In conclusion, increasing travel speed and cooling time significantly affects pore diameter due to the lower heat input to the weld zone. Porosity can be reduced when diminishing heat accumulation. The results indicate that an increase in travel speed produces a slight decrease in porosity. Specifically, the total pore volume diminishes from 0.42 to 0.36 mm^3^ when increasing the travel speed from 700 to 950 mm/min. The ultimate tensile strength and maximum elongation of the ‘back and forth’ strategy are slightly higher than those of the ‘go’ strategy. After tensile testing, the ultimate tensile strength and yield strength did not show any relation to the porosity measured by computed tomography. The percentage of the pore total volume over the measured volume was lower than 0.12% for all the scanned specimens.

## 1. Introduction

Additive manufacturing is a key technology in the 4th Industrial Revolution and has garnered significant attention from researchers and industry in recent years [1]. The additive manufacturing processes provide this new industrial context with a technology suitable for obtaining customized products at the point of application, reducing waste through optimized material utilization [2].

There are seven classes of additive manufacturing processes according to ISO 52900 [3]. Among them, the processes that enable the manufacture of metallic materials are directed energy deposition, powder bed fusion, and sheet lamination. Directed energy deposition offers benefits such as the ability to produce large parts, the capability to work on both uniform and irregular surfaces, and the option to combine materials. Wire arc additive manufacturing (WAAM) is one of the main directed energy deposition processes, allowing the manufacture of metal parts in aluminum, nickel, steel, and titanium alloys [4,5,6]. Moreover, it offers additional advantages such as low cost of facilities and equipment, and no need for vacuum creation as required in electron beam-based processes [7].

The high local heat input and high material deposition rate are important characteristics of WAAM [8]. The accumulation of heat in successive layers is critical. Zhang et al. [9] investigated the heat transfer control system to evaluate the solder droplets frequency and size and, lastly, to improve precision. This study tried to suppress imperfections such as porosities and coarse microstructures by developing a new approach, which employed wire arc additive manufacturing enhanced by workpiece vibration with a variable polarity cold metal transfer arc, employing high-strength aluminum alloy. Cold metal transfer (CMT) provides excellent bead quality, low heat input, minimal spatter, high cladding efficiency, significant material waste reduction, and high-energy efficiency [10]. Chen et al. [11] studied the stability of aluminum alloy formation using the WAAM-CMT system by altering process parameters, robot paths, and Fronius digital welding modes. They concluded that the curvature of the surface of the parts affects the thickness and layer height. A higher curvature increases the variance value of the width and size of the deposited layer, and negatively affects the stability of the process. Yang et al. [12] proposed using the double electrode–gas metal arc welding (DE-GMAW) process to minimize energy transfer to the base metal. Pereira et al. [13] studied the robotic WAAM process to analyze the geometry and surface topography of inclined walls made of AWS A5.18. ER70S-6 steel. They found that travel speed affected the cross-section size due to the heat input. Moreover, the researchers identified the crucial role of intermediate cooling to obtain structures with more uniform dimensions. Chaudari et al. [14] obtained multi-wall structures of 1.25Cr-1.0Mo steel using WAAM technology. The researchers identified how the tensile properties were similar to those of 1.25Cr-0.5Mo metal-cored wire [14]. Similarly, Vora et al. [15] identified how the WAAM process also allows obtaining tensile properties close to the range of wrought SS 316 L.

The main disadvantages of the WAAM process are the need for substrate material to deposit the material, the lack of dimensional precision, and the poor surface quality [11]. Moreover, the appearance of porosity might decrease the mechanical properties of the parts [16,17]. Several mechanisms can produce porosity. For instance, hydrocarbon contaminants on the metal wire surface can cause atomic hydrogen to be absorbed directly into the molten pool. Micropores may appear during shrinkage due to incomplete solidification. The state of the wire is therefore critical as moisture, grease and hydrocarbon contaminants on the wire surface can be vaporized in the arc. Moreover, unstable processes and poor process parameter selection can lead to the appearance of porosity and voids [18]. In addition, harmful residual stresses may appear, modifying the geometry of the parts. Although a proper selection of the welding conditions to minimize the energy input through CMT welding systems can decrease the appearance of residual stresses [19], the manufactured parts usually require machining to meet geometry and surface finish requirements.

Today, coordinate measuring systems perform nearly all modern/advanced geometric measurements. Computed tomography (CT) allows the measure of surfaces that cannot be accessed by contact or optical devices, and to evaluate the object’s internal structure. This includes determining the presence of porosity or the distribution of fibers in plastics. Technical tomography differs from medical one in the higher resolution of the image obtained and the power of the X-ray tube. In the end, the result is a reconstruction of X-ray images. The lamp and detector are often stationary in technical tomography, and the measured object is rotated 360° on a rotary table. This rotation is divided into several hundred to several thousand positions, for which the object’s X-rays are recorded. In places where the object is characterized by a higher density and/or thicker cross-section, a more substantial attenuation of radiation occurs, as shown by darker areas of the X-ray image. The next step is the reconstruction based on Feldkamp-Davis-Kress algorithms (FDK algorithms) [20,21]. These calculations result in creating a volumetric image, allowing for dimensional analysis, internal structure evaluation, or simulation studies.

Micro-computed tomography (micro-CT) is an increasingly popular technique for geometric measurements [22]. It is beneficial for measuring parts with complex shapes and defined internal structures, often manufactured with additive techniques [23]. It is a non-destructive test, which allows volumetric evaluation of, for example, porosity, not only in terms of percentage but also in terms of pore location and size [24]. The most common cavity defect related to WAAM of aluminum is hydrogen porosity. This defect is spherical and formed due to supersaturated hydrogen precipitation during solidification [25]. Recently, it has also become increasingly popular to use surface topography measurements [26], allowing for the examination of re-entrant surfaces.

The 5000 series aluminum alloys have magnesium as the primary alloying element. They include small amounts of manganese to enhance their strength, and magnesium to increase corrosion resistance, which makes this alloy especially suited for marine environments [27]. In recent years, there has been an increase in the research and application of WAAM for the 5000 series alloys such as the 5183 [28,29] and 5356 [30,31,32] aluminum alloys. For instance, Derekar et al. [33] investigated the effects of changing the cooling time on the porosity and mechanical properties of WAAM parts prepared using a DC pulsed GMAW process with 5356 aluminum consumable wire. The researchers used the micro-CT technique to study the distribution of pores in tensile test specimens manufactured by WAAM, testing different process parameters.

Even though CT is increasing its share in manufacturing research, the technique has still not been widely used in additive manufacturing. CT may still play a major role in better understanding the layer-by-layer processes and, specifically, in manufacturing metallic parts by WAAM. Understanding the influence of the manufacturing parameters on defects such as porosity may help reduce them and, thus, obtain enhanced parts.

This paper aims to provide the reader with an experimental study on selecting WAAM parameters for creating tensile testing specimens of the 5356 aluminum alloy. The study considers cooling time, travel speed and path strategy and their influence on the porosity occurrence measured by CT. Moreover, the relationships between porosity and mechanical properties are also explored.

## 2. Materials and Methods

This section presents all details regarding the experimental arrangements. First, the materials are presented. Second, the equipment used for welding is described. Third, the research methodology used to study the WAAM process is introduced. Fourth, the design of the specimen used for testing is discussed. Fifth, the measuring procedures used to evaluate the WAAM process are presented.

### 2.1. Materials

The welding wire material used is a 5356 aluminum alloy (AWS–ER5356) with a diameter of 1 mm, supplied in 6 kg coils by Pastoriza S.L. The alloy provides good weldability and corrosion resistance and finds application in locomotive carriages, chemical pressure vessels, ships, and aviation [34]. The technical specifications for the material are detailed in Table 1 and Table 2 [35]. As shown, the alloy has 5% magnesium as the primary alloying element to increase corrosion resistance and also includes small amounts of manganese (0.05%) to enhance its strength. The substrate metal, also alloy 5356, is in the form of 5 mm thick sheets. An argon gas (99.9%) was used as a protecting gas according to ISO 14175/I [36]. Argon gives better results than nitrogen, according to Li et al. [12].

### 2.2. Equipment

The experimental equipment used in WAAM manufacturing was composed of two different systems: The welding machine was a Fronius TPS 4000 CMT R machine, which allows welding using Fronius^®^ patented CMT technology. The results are reduced welding temperature and optimized wire movement. Thus, the machine offers a better weld seam quality than the conventional GMAW welding process.Positioning system. The BF 30 Vario Optimum CNC milling machine efficiently manages the movement of the entire system. This machine was adapted to position the torch weld in the Z axis. The movement of the X-Y table of the CNC system makes it possible to deposit a layer of weld on a fixed Z level with the welding torch attached to the milling machine head. An auxiliary worktable was required to isolate both systems electrically.

Figure 1 depicts the experimental setup described in this section [37].

### 2.3. Reseach Methodology

Figure 2 depicts the methodology employed in the study. First, an experimental plan defines the order of the experiments and combinations of the variable factors (travel speed, cooling time, and path strategy) to evaluate the influence of the selected manufacturing parameters on optimizing mechanical characteristics. This stage helps to define the conditions for creating the walls. Previously, some testing was needed to find suitable welding parameters for the experiment. Second, the design of walls and programming are performed. Good specimen design is required to obtain tensile testing specimens from the walls. Third, walls are manufactured by WAAM, and the specimens are cut out from the walls. Fourth, to study the influence of manufacturing parameters and mechanical characteristics, the porosity of the specimens was measured by computed tomography. Then, tensile testing has been performed with the WAAM specimens.

Linear wall samples were fabricated by using CMT and the pulsed method. The penetration and weldability are related to travel speed and the material’s conductivity [38]. The selection of the WAAM parameters must result in adequate heat input for the material to be welded. In this sense, Su et al. [39] worked with approximate heat input in the range 140–240 J/mm in CMT WAAM of the 5356 alloy. The selection of the wire feed and travel speed is relevant to the process. In this sense, Aldalur et al. [40] tested values in the 400–800 mm/min and 330–1000 mm/min ranges, respectively. 

Gas is used in the GMAW process to protect against forming an aluminum oxide scale [41]. Despite the beneficial effect of the protective gas, the flow rate may influence the appearance of porosity, as Hauser et al. [42] claimed. 

Based on the previous and pre-testing trials, the manufacturing parameters were selected. These WAAM parameters were a current of 70 A, a voltage of 14.7 V, a wire feed of 6.5 m/min, pulsed CMT mode, with an argon gas flow rate of 9 L/min (see Table 3). The selection of 32 layers allowed for an average height wall of 35 mm to be obtained.

The heat input is a critical factor in the process [4,43]. Equation (1) allows the calculation of the heat input using the travel speed (*S*), average arc current (*I*), arc voltage (*V*), and process efficiency (*η*). The process efficiency ranges from 0.8 to 0.9 [40,44], assuming it is 0.8 for CMT in this study [45].
(1)$$HI\;(kJ/mm)=\frac{\eta *V\;\left(V\right)*I(A)}{1000\;(J/kJ)*S\;(mm/s)}$$

The selected varying parameters were the travel speed, the cooling time, and the path strategy. Three levels of travel speed were chosen (700, 825, and 950 mm/min). Two levels were selected for the cooling time (60 to 30 s) and path strategy (‘Go’ and ‘Back and forth’). The design of experiments was created by fixing one parameter and varying the two others using two levels. In this sense, A, E, and V tests were performed at a fixed travel speed of 925 mm/min, a cooling time of 60 s and a ‘Back and forth’ path strategy, respectively [13]. These testing conditions are displayed in Table 4. In addition, two replicates of the testing conditions were performed. Thus, 36 walls were created.

### 2.4. Design and Manufacturing of the Specimens

From the 36 walls, 24 tensile testing specimens were manufactured. In this sense, the best two specimens of each of the 12 combinations were selected based on visual inspection. The tensile testing specimens were manufactured by machining on a Microcut with the Fagor 8065 CNC software. The insert tool was XNGM120308. It was necessary to design and develop an adequate clamping device. Figure 3 shows the layout of the tensile testing specimens. Anisotropy is critical in layer-by-layer processes [46]. In this case, all specimens were cut in the same direction, as shown in Figure 3.

### 2.5. Measurements

Two tests were carried out to study the 24 tensile testing specimens: computed tomography measurements and mechanical testing. Thus, firstly, an analysis by computed tomography of the specimens was performed to obtain the following values: mean pore diameter, standard deviation of the pore diameter, mean pore volume, total pore volume, and standard deviation of pore volume.

The measurements were conducted in a CT GE PhoenixV|tome|XS in the Metrology Laboratory of Poznan University of Technology, as shown in Figure 4. The calibration of the system and its construction give the possibility to measure porosity. The conditions of measurements consist of minimal voxel side equal to 26 µm, a sample size of 45 mm × 15 mm, an image width of 690 pixels, scanning with 1000 images and 360° per sample, a voltage of 90 kV, a current of 220 µA, and an exposure time of 200 ms [47].

The second test was tensile testing to obtain the maximum force, maximum elongation, yield strength, and ultimate tensile strength. The tensile testing was conducted using a 50 kN Hegewald&Peschke Universal Machine at ITA company, according to DIN EN ISO 7500-1 [48], as shown in Figure 5. 

## 3. Results and Discussion

The computed tomography measurements of the specimens are displayed in Table 5. The table includes the codes of the selected specimens, varying parameters, and porosity measurements. Specifically, mean pore diameter, mean pore volume, standard deviation of pore diameter, standard deviation of pore volume, and pore total volume are listed in Table 5.

Moreover, Table 6 displays the tensile testing results ordered by cross-section.

### 3.1. Process Parameters and Porosity Analysis

Figure 6 shows the pore diameter and its standard deviation versus the travel speed and cooling time. It is well known that the higher the travel speed, the lower the heat input (Equation (1)). Köhler et al. [31] investigated the heat accumulation and the influence of different temperature-time regimes on the resulting component properties during WAAM of the 5356 aluminum alloy. The authors highlighted that heat accumulation changed with travel speed and cooling time. Reducing the heat input can improve the microstructure and reduce the porosity of aluminum alloy components. 

Regarding the average pore diameter, the higher the travel speed, the lower the pore diameter. As the cooling time increases, the size of the pore diminishes. The optimum setup must include higher travel speeds and longer cooling times to reduce pore diameter. Thus, the size of the pores gets smaller when heat input and heat accumulation are lower [6,49,50]. In addition, using CMT based WAAM allowed controlling porosity as indicated by Fang et al. [51] when using the 2219 alloy. The authors obtained pore sizes under 100 µm. Moreover, using a gas flow rate of 9 L/min might help reduce porosity. Hauser et al. [42] claimed that using high flow rates produces rapid solidification of the melt pool by forced convection. The present results agree well with the study by Zhou et al. [52] when using travel speeds from 150 to 450 mm/min. As the cooling time increases, the size of the pores diminishes. Less porosity was found by Derekar et al. [33] when increasing the interpass time. 

When computing the total volume occupied by the pores, the influence of the travel speed is evident. Thus, Figure 7 shows the behavior of pore total volume versus the travel speed. The total pore volume decreases from 0.42 to 0.36 mm^3^ when increasing the travel speed from 700 to 950 mm/min.

### 3.2. Relationship of Process Parameters and Tensile Test 

As shown by the tensile test results in Table 6, significant differences in thickness and length were obtained depending on process conditions. This means that the cross-section of the machined specimens was not uniform. The results listed in Table 6 include maximum force, ultimate tensile strength, and yield strength. These results are similar to those reported by Haselhuhn et al. [53] and slightly lower than those with micropulsed MIG welding studied by Jiangang et al. [34]. Jiangang et al. [34] reported that no significant cracks were found in WAAM specimens with minor porosity, and the aggregation of equiaxed grains improved the mechanical properties. The authors measured ultimate tensile strength slightly higher than other studies and the as-cast 5356 aluminum alloy. Su et al. [40] reported ultimate tensile strength values of 255 ± 5 MPa. The present study’s range for ultimate tensile strength is higher (230.57–274.88 MPa).

Several specimens broke close to the clamping zone, which might indicate that it is outside the scope of the CT scanner’s capabilities. This suggests that CT results are unrelated to the tensile test results. By analyzing the results some outliers can be identified as marked in Table 6 (V1.3, A1.1, E4.2, E2.1 and A2.2).

The A1.1 and V1.3 specimens showed close maximum forces (10.72 and 10.66 kN, respectively). These results show no relation with the pore total volumes of 0.51 and 0.14 mm^3^, respectively (Table 5). However, it should be noted that the breakage of the A1.1 specimen (Figure 8) occurred almost outside the scanned area. Therefore, the ultimate tensile strength might be unreasonable for a medium-high total volume of pores. However, the V1.3 specimen has the breakage zone centered, which has been scanned and has a reasonably small pore total volume.

Regarding the E4.2 and E2.1 excluded values, it is worth mentioning that E4.2 and E2.1 specimens have high mean diameters of porosity of 0.43 and 0.45 mm, respectively. The average diameter of the pores of the A2.2 specimen was slightly lower (0.37 mm). However, it should also be considered abnormal because the breakage did not occur in the middle section.

Concerning the path strategy, Figure 9 shows minimal variation between the ‘Back and forth’ strategy and the ultimate tensile strength. In the case of the ‘Go’ strategy, a slight difference is noticeable, although it does not exceed a 3.5% deviation. The ultimate tensile strength was higher in the ‘back and forth’ strategy. The difference is more significant for the higher travel speed (‘Back and forth’) and medium travel speed (‘Go’), reaching an increase of the ultimate tensile strength of 5%. 

The relationship between maximum elongation and path strategy is shown in Figure 10. The strategy ‘Back and forth’ is better for increasing the maximum elongation, especially when the cooling time was of 30 s. 

Figure 11 shows the 3D CT, the stress-strain graph, and the specimen after testing. These results correspond to the best case (V1.3, with maximum ultimate tensile strength) and the worst case (E2.2, with minimum ultimate tensile strength). It was observed that V1.3 had a total volume of pores (0.14 mm^3^) close to the minimum value (0.11 mm^3^). On the other hand, in the case of the E2.2 specimen, the value of 0.24 mm^3^ was not the highest value. The highest value corresponds to the V3.3 specimen, with a value of 0.94 mm^3^.

### 3.3. Relationship of Porosity and Tensile Test Parameters

Figure 12 shows the ultimate tensile strength and yield strength versus the pore total volume. There is no direct correlation, contrary to what might be expected. Thus, it seems that the volume of the pores is not high enough to affect the mechanical properties. 

Figure 13 shows the relationship between the ultimate tensile strength and yield strength versus the percentage of the pore total volume over the total measured volume. The highest percentages of pores do not exceed 0.12%. It is interesting to highlight that, in this case, the value of the ultimate tensile strength is high (259 MPa). In addition, the random behavior of the ultimate tensile strength concerning the percentage of pore total volume over measured volume indicates that the ultimate tensile strength was not affected by these porosity levels. Similarly, the yield strength did not show any clear relation to the porosity.

## 4. Conclusions

The present study analyzed the manufacturing of walls created with the WAAM process. The 5356 aluminum alloy was used to build walls that were later machined to obtain tensile testing specimens. These specimens were analyzed by computed tomography and then mechanically tested. The following conclusions can be drawn from the experimental study:The diameter of the pores slightly decreases with the increment of travel speed and cooling time. Thus, lower heat accumulation provided a positive outcome in terms of porosity. Specifically, the total pore volume decreases from 0.42 to 0.36 mm^3^, i.e., it decreases by 14% when increasing the travel speed from 700 to 950 mm/min.The WAAM manufacturing process allowed for obtaining walls with high ultimate tensile strength.There is almost no variation of ultimate tensile strength if the travel speed changes, but the best strategy was ‘Back and forth’. The maximum elongation was also higher when using the ‘back and forth’ strategy.The ultimate tensile strength and the yield strength were unrelated to the pore total volume. It should be highlighted that the percentage of the pore total volume over the measured volume was lower than 0.12% for all the scanned specimens.

The present study is an initial experimental evaluation of CMT-based WAAM of 5356 aluminum alloy. The weldability of aluminum alloys is challenging and a proper selection of WAAM parameters is needed. Therefore, future work will optimize these parameters based on heat input and heat accumulation. Thus, proper temperature control during the tests is of relevance. Microstructure was not analyzed in this study. Thus, it is another crucial issue in new experiments. In addition, the analysis of porosity can be conducted in more detail. An understanding of the distribution of porosity may be helpful. Finally, the flow rate of the protective gas was selected at a low value (9 L/min). In this sense, it would be interesting to vary this parameter and see how it relates to results other than the porosity.

## Figures and Tables

**Figure 1 materials-16-02570-f001:**
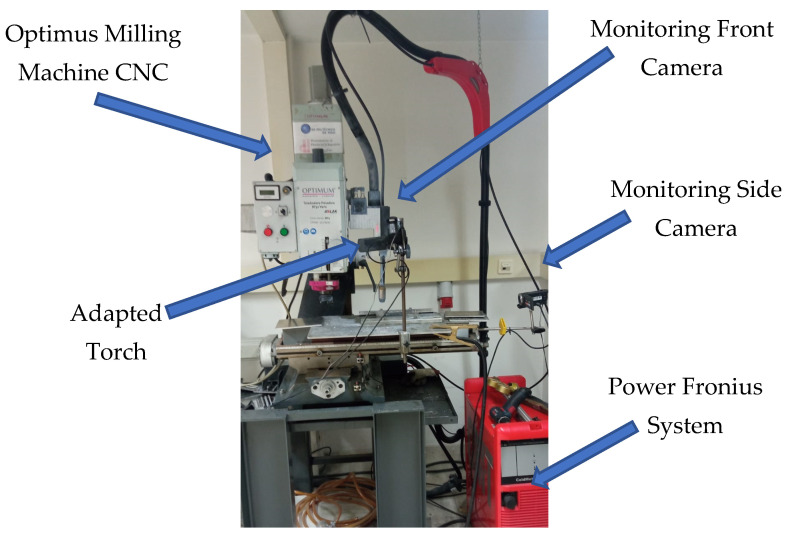
Experimental integration of the CNC systems Optimus CNC and Fronius CMT.

**Figure 2 materials-16-02570-f002:**
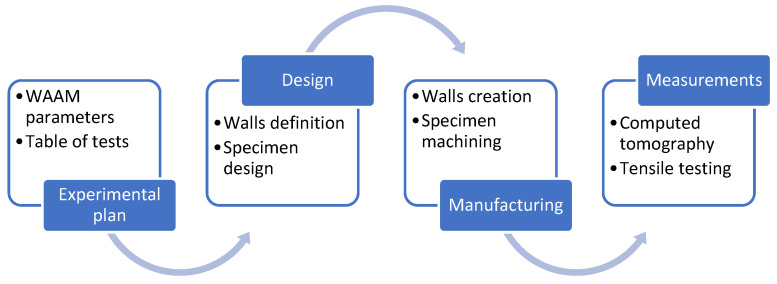
Research methodology.

**Figure 3 materials-16-02570-f003:**
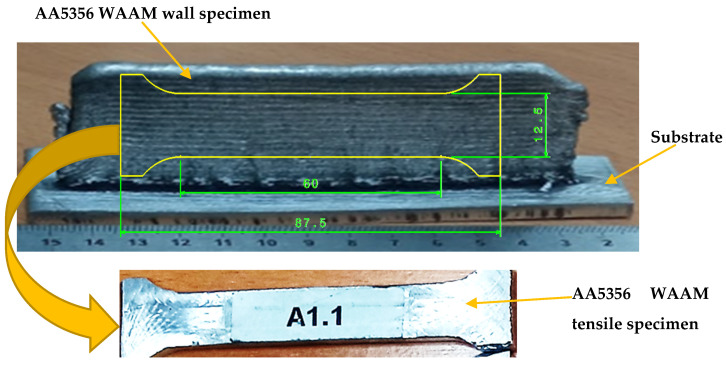
Tensile specimen layout according EN ISO 6892-1:2016.

**Figure 4 materials-16-02570-f004:**
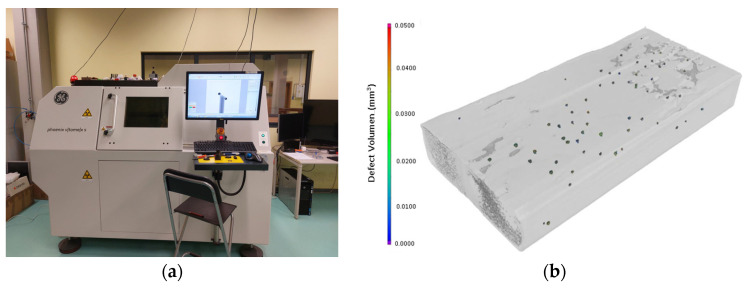
(**a**) CT GE Phoenix v|tome|x S in the Metrology Laboratory of Poznan University of Technology, (**b**) Porosity analysis of the A1.2 specimen.

**Figure 5 materials-16-02570-f005:**
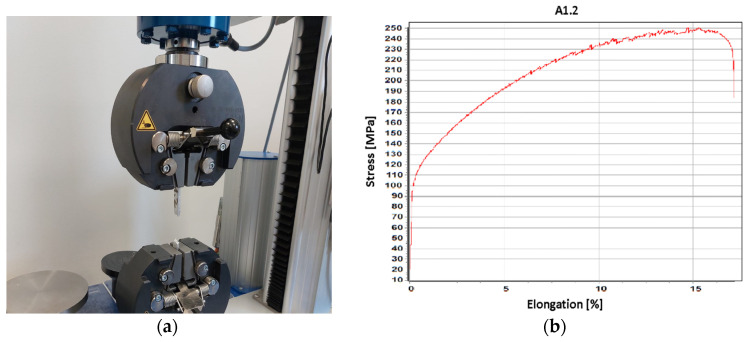
(**a**) Tensile test on Hegenwald&Perschke Universal Machine (**b**) Stress-Elongation graph of the A1.2 specimen.

**Figure 6 materials-16-02570-f006:**
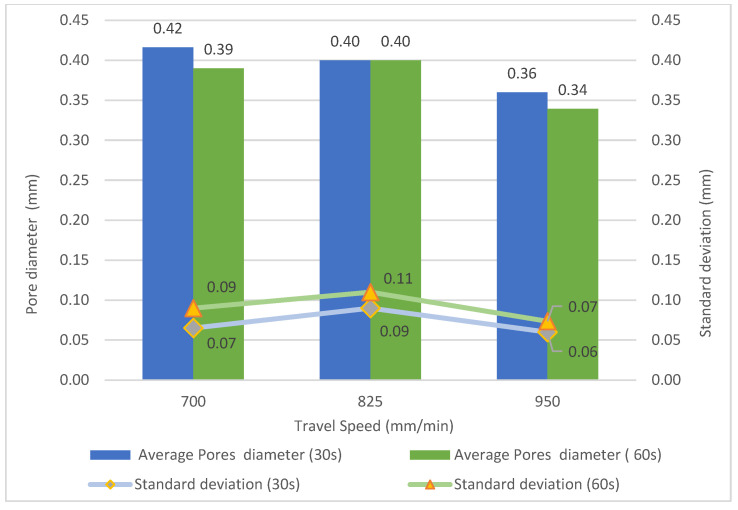
Mean pore diameter and its standard deviation versus travel speed for cooling times of 30 and 60 s.

**Figure 7 materials-16-02570-f007:**
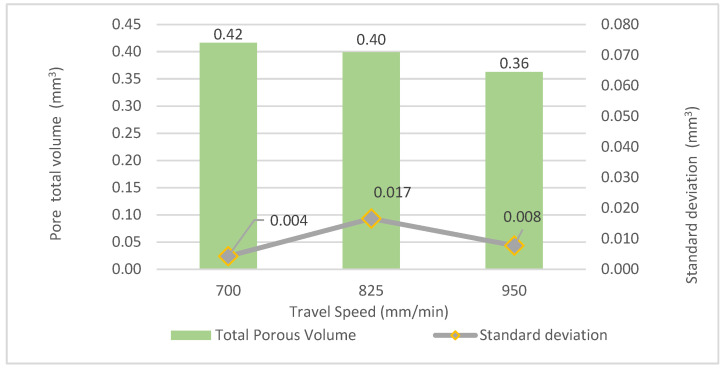
Pore total volume versus travel speed.

**Figure 8 materials-16-02570-f008:**
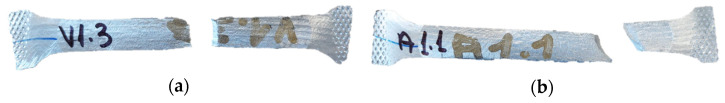
Specimens after tensile testing: (**a**) V1.3; (**b**) A1.1.

**Figure 9 materials-16-02570-f009:**
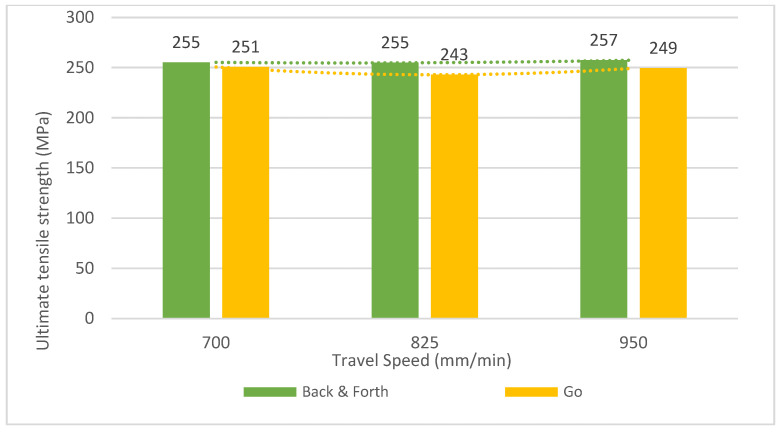
Ultimate tensile strength versus travel speed and path strategy.

**Figure 10 materials-16-02570-f010:**
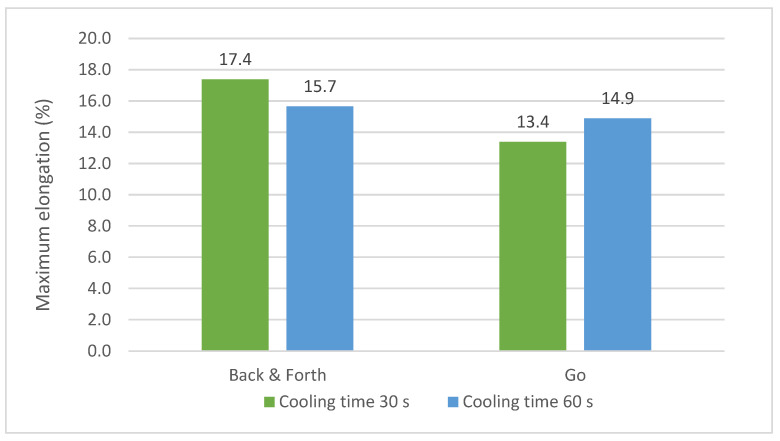
Max. elongation versus path strategy and cooling time.

**Figure 11 materials-16-02570-f011:**
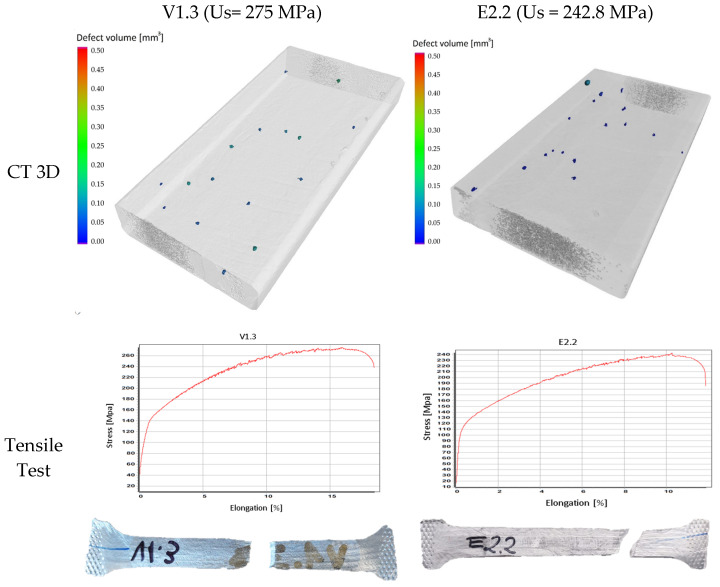
Graphical results of CT and tensile testing of V1.3 and E2.2 specimens.

**Figure 12 materials-16-02570-f012:**
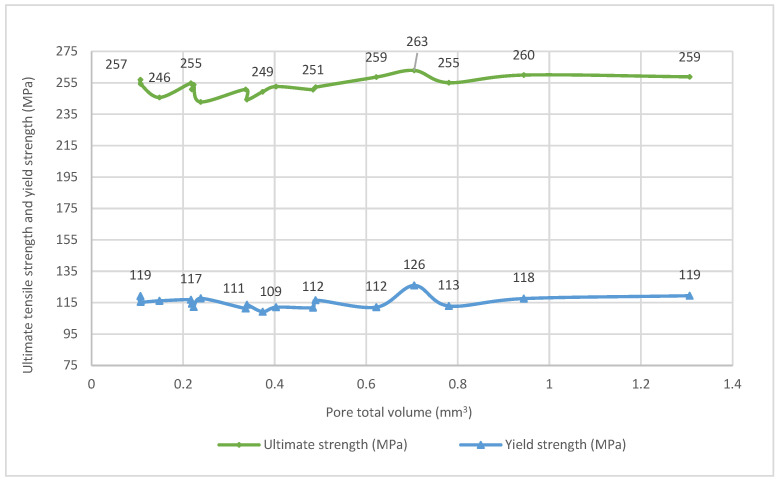
Ultimate tensile strength and yield strength versus the pore total volume.

**Figure 13 materials-16-02570-f013:**
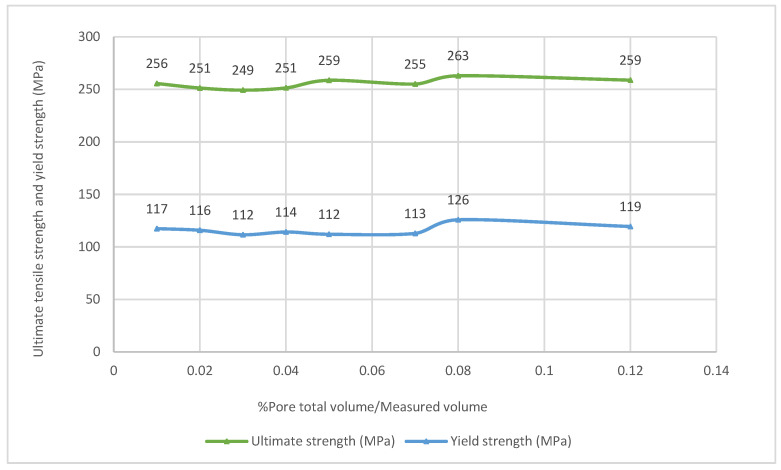
Ultimate tensile strength and yield strength versus the percentage pore total volume/measured volume.

**Table 1 materials-16-02570-t001:** Chemical composition of the 5356 aluminum alloy (%).

Mg	Fe	Si	Cu	Sn	Ti	Cr	Mn	Be	Zn
5.0	0.40	0.25	0.10	-	1.10	0.2	0.05	-	0.10

**Table 2 materials-16-02570-t002:** Properties of the 5356 aluminum alloy at 25 °C.

Characteristic	Value
Density, ρ (kg/m^3^)	2640
Specific heat capacity, c (J/kg·K)	900
Thermal conductivity, k (W/m·K)	121.8
Coefficient of thermal expansion, α (°C^−1^)	22.5
Modulus of elasticity, E(GPa)	70
Poissons ratio, µ	0.33
Yield strength, σy (MPa)	130

**Table 3 materials-16-02570-t003:** Fixed WAAM parameters.

WAAM Parameter	Value
Current (A)	70
Voltage (V)	14.7
Wire feed speed (m/min)	6.5
Mode1	CMT
Mode2	S2
Gas	99.9% Ar
Gas flow rate (L/min)	9

**Table 4 materials-16-02570-t004:** Design of experiments for wall manufacturing.

Test	Walls	Travel Speed (mm/min)	Cooling Time (s)	Path Strategy	Heat Input (kJ/mm)
A1	A1.1/A1.2/A1.3	950	30	Back & Forth	0.052
A2	A2.1/A2.2/A2.3	950	30	Go	0.052
A3	A3.1/A3.2/A3.3	950	60	Back & Forth	0.052
A4	A4.1/A4.2/A4.3	950	60	Go	0.052
E1	E1.1/E1.2/E1.3	825	60	Back & Forth	0.060
E2	E2.1/E2.2/E2.3	825	60	Go	0.060
E3	E3.1/E3.2/E3.3	700	60	Back & Forth	0.071
E4	E4.1/E4.2/E4.3	700	60	Go	0.071
V1	V1.1/V1.2/V1.3	700	30	Back & Forth	0.071
V2	V2.1/V2.2/V2.3	700	60	Back & Forth	0.071
V3	V3.1/V3.2/V3.3	825	30	Back & Forth	0.060
V4	V4.1/V4.2/V4.3	825	60	Back & Forth	0.060

**Table 5 materials-16-02570-t005:** Results of tomography measurements of tensile testing specimens.

Specimen	Mean Pore Diameter (mm)	Mean Pore Volume (mm^3^)	Standard Deviation of Pore Diameter (mm)	Standard Deviation of Pore Volume (mm^3^)	Pore Total Volume (mm^3^)
A1.1	0.43	0.014	0.117	0.039	0.51
A1.2	0.33	0.006	0.064	0.003	0.34
A2.2	0.37	0.006	0.054	0.002	0.23
A2.3	0.39	0.007	0.067	0.003	0.34
A3.1	0.34	0.007	0.097	0.007	0.70
A3.2	0.35	0.006	0.069	0.003	0.78
A4.1	0.33	0.006	0.071	0.003	0.40
A4.3	0.34	0.006	0.057	0.002	0.22
E1.2	0.36	0.012	0.191	0.033	0.22
E1.3	0.37	0.015	0.160	0.029	0.49
E2.1	0.45	0.009	0.091	0.003	0.16
E2.2	0.38	0.013	0.120	0.023	0.24
E3.1	0.39	0.007	0.060	0.002	0.70
E3.2	0.32	0.006	0.099	0.003	0.31
E4.1	0.38	0.008	0.060	0.005	0.48
E4.2	0.43	0.010	0.123	0.008	0.39
V1.2	0.42	0.008	0.056	0.003	0.11
V1.3	0.41	0.009	0.074	0.004	0.14
V2.2	0.41	0.010	0.095	0.007	0.15
V2.3	0.41	0.007	0.080	0.003	0.11
V3.2	0.40	0.007	0.094	0.003	0.62
V3.3	0.39	0.010	0.143	0.033	0.94
V4.1	0.40	0.008	0.072	0.006	0.37
V4.2	0.40	0.007	0.085	0.003	0.22

**Table 6 materials-16-02570-t006:** Tensile testing results, ordered by section (mm^2^).

Specimen	So (mm^2^)	Fmax (kN)	Ultimate Tensile Strength (MPa)	Max Elongation (%)	Yield Strength (MPa)
A3.1	30.00	7.98	266.00	11.60	134.04
A4.3	35.04	8.79	250.86	15.29	114.38
E2.2	36.67	8.91	242.98	11.86	117.80
V2.2	37.5	9.22	245.87	12.73	116.22
(*) V1.3	38.78	10.66	274.88	18.48	133.44
E3.2	39.38	10.19	258.76	14.91	119.44
(*) A1.1	39.66	10.72	270.30	18.21	129.92
E3.1	39.69	10.31	259.76	17.49	117.87
V1.2	40.13	10.21	254.42	15.31	115.58
(*) E4.2	40.91	9.57	233.93	11.14	115.28
(*) E2.1	41.01	9.56	233.11	13.25	109.75
E4.1	41.27	10.34	250.55	15.56	111.91
A2.3	42.68	10.43	244.38	13.38	113.75
V2.3	42.83	11.01	257.06	14.55	119.24
(*) A2.2	42.85	9.88	230.57	14.89	116.70
E1.2	42.89	10.93	254.84	18.25	116.94
A3.2	43.46	11.09	255.18	15.12	112.91
V3.3	44.25	11.50	259.89	18.61	117.64
V3.2	44.50	11.51	249.21	17.08	109.26
V4.1	44.50	11.09	258.65	18.42	112.14
A4.1	44.82	11.32	252.57	14.46	112.13
E1.3	45.52	11.48	252.20	16.60	116.59
V4.2	45.85	11.64	253.87	17.62	112.27
A1.2	47.10	11.81	250.74	17.18	111.35

(*) anomalous values.

## Data Availability

Data are contained within the article.

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
