# Peer review of "Characterization of 5356 Aluminum Walls Produced by Wire Arc Additive Manufacturing (WAAM)"

_materials, 2023, doi:10.3390/ma16072570_

Round 1
Reviewer 1 Report
This manuscript conducted systematic work on the effect of WAAM parameters on Al formation. Overall this manuscript is well-written and the validity and robustness of the method are clear. So I would recommend major revision if the author could address the following questions and further refine their work.
1. The reduction of the travel speed will increase heat accumulation and thus increase porosity. How to solve this dilemma?
2. What about the surface roughness of the component?
3. The author mentioned lots of methods to reduce porosity. It is good. But can you elaborate on what are the physical origin of the porosity?
4. After the deposition, is there any composition change of the 5356 aluminum alloy?
5. According to the minimum voxel, the step size of CT is around 100 um. Is it sufficient to characterize the smaller porosities?
6. I suggest the author re-draw the SS curve in Figure 5. It should be clearer.
7. In Line 224, I do not understand why higher speed leads to lower porous diameter. The author mentioned that cooling time positively correlates with porous size. this seems to be contradictory. What is the relationship between the cooling rate and porous size? Please explain.
8. And also, please explain why?
Author Response
Helllo, our reply is in the attached file
Best regards
Michal Wieczorowski

Reviewer 2 Report
In the manuscript “Characterization of 5356 aluminum walls produced by wire arc additive manufacturing (WAAM)”, authors used WAAM process with a cold metal transfer (CMT) Fronius system. It contains some interesting key findings and it will be useful for researchers working in this domain. However, I think that this manuscript requires major improvements in following areas:
- Add Multi-layered walls as keyword
- Abstract is not written well. Add more qualitative and quantitative findings. It is completely missing in the abstract.
- I didn’t observed any information about the selection of work material in the introduction section. It must be added.
- Lot of abbreviations were used in the present study. Describe them in their first appearance throughout the manuscript.
- Add more literature of recent work carried out by the researchers and clearly mention the research gap through that study. Following articles will be useful for this purpose: https://doi.org/10.1016/j.jmrt.2022.08.074; https://doi.org/10.1016/j.jmrt.2022.10.158
- Avoid having one heading after another with no discussion in between as in the case of Sections 2, and 2.1
- On what basis the parameter values were selected??
- Line 182-185: rephrase the sentence.
- Figure 4 a: Scale bar values are not visible
- Results are very interesting. However, it is necessary to give technical reasons behind each and every findings. This part is missing in the discussion section.
- Revise the conclusion section. Add quantitative findings as well.
- Mention the limitations and further scope of improvement in last section.
Author Response

(The authors gave the same response as above.)

Reviewer 3 Report
The article investigated the effect of WAAM parameters on the porosity and tensile strength of an aluminum alloy. The article has a scientific novelty and may be useful to other researchers. The article can be published in a high-ranking scientific journal, but it needs to be corrected. Here are my notes that may help authors improve the quality of the paper.
1. Figure 3 must be redrawn in a graphical editor with all dimensions indicated. What is the standard for the geometry of a tensile test specimen? Provide a photo of the original sample from which the test sample was cut. Give a more detailed layout of the test samples in the original sample. How are the wire fusion tracks located?
2. In figures 4 and 11, the scale is too small and has poor resolution. The numbers and colors of the pores are indistinguishable. The drawings need to be significantly improved, because now they do not carry information.
3. In table 5 in column 1 there is a designation of samples, what do these letters (A, E, V) and numbers (1.1; 1.2 ... 4.2) mean? What does "path strategies ‘Go’, ‘Back and forth’" mean?
4. The graphs in Figures 12 and 13 are poorly designed. Numbers and words are mixed. What is the need to indicate the values ​​of tensile strength and yield strength with two signs after the "." ?
5. Why don't you analyze the elongations after having the results of the tensile tests?
6. Conclusions need to be reworked. Now they do not have numerical parameters, but are of a general nature. Indicate the quantitative values obtained as a result of the research.
7. What is the scientific meaning of Figure 8?
8. Add a discussion of the results. Explain the reasons for your results. Now the article looks like an unfinished paper that needs to be finalized .
Author Response

(The authors gave the same response as above.)

Reviewer 4 Report
The authors of this paper presented the findings of their investigation into analyzing the specimens’ porosity and its potential relation to the tensile strength of 5356 aluminum walls produced by WAAM. Some tests were performed, including computed tomography (CT) and tensile test. In addition, the variation of some of this additive manufacturing parameters (travel speed, intermediate cooling, and path strategy) was also examined. The study has been structured well. However, given the remarks below, the manuscript needs to be revised.
- Table 2: Even if the characteristic symbols (of 5356 aluminum alloy listed in Table 2) are common in the field, it is better to define what does it mean each one of them.
- Line 189: Since authors are talking about the measurements which were conducted in a CT, so they mean Figure 4.a rather than Figure 3.a.
- Line 196: Authors referred to the tensile testing universal machine, therefore, Figure 5 should be mentioned instead of Figure 4.
- Line 254: Authors say "has a reasonably small TPV value.", I guess they mean PTV (Porous Total Volume) rather than TPV. Please check it.
- Figure 10: What could be the explanation for the significant difference in the ultimate strength of Go at 825 travel speed compared to other travel speeds tested?
- Figure 12: The cubic in the unit of the x-axis should be superscript (i.e., mm3 or mm^3).
- Line 295: Correct the typo "tomogra-phy" please. Also, in line 296, Correct "ex-perimental".
- Conclusions: The second point (line 302) is just repeating the first point, as the diameter and volume of the porous are related to each other. So, no need to mention them twice.

Author Response

(The authors gave the same response as above.)

Round 2
Reviewer 1 Report
The authors have made substantial improvements in this version. So it is qualified for publishing.
Reviewer 2 Report
Accept in present form
Reviewer 3 Report
The authors have substantially corrected the article. I recommend it for publication in this version.